# SQS: Speech Quality Assessment in the Data Annotation Context

## Abstract

Audio quality plays a crucial role in the data annotation process as it influences various factors that could significantly impact the annotation results. These factors include transcription speed, annotation confidence, and the number of audio replays, among others. Consequently, transcriptions often contain numerous errors and may have blank or incomprehensible sections. Most existing objective measures (e.g., Perceptual Evaluation Score Quality (PESQ), Speech Intelligibility Index (SII)) and subjective measures (e.g., Mean Opinion Score (MOS)), and speech quality measures (e.g., Word Error Rate (WER)) do not consider factors that could hinder the annotation process. These measures poorly correlate with the audio quality perceived by the annotator in the annotation context. We propose a novel subjective speech quality measure within the audio annotation framework, called Speech Quality Score (SQS). This measure encompasses the most relevant characteristics that can impact transcription performance and, consequently, annotation quality. Additionally, we propose a DNN-based model to predict the SQS measure. Our experiments were conducted on a dataset composed of 1,020 audio samples with SQS annotations created specifically for this study, using the RTVE2020 Database. The results demonstrate that our proposed model achieved high performance with a linear correlation coefficient of 0.8 between ground-truth and predicted SQS values. In contrast, state-of-the-art MOS prediction models exhibited a poor correlation (i.e., 0.2) with ground-truth SQS values.

## 1 Introduction

Speech intelligibility is directly associated with audio quality. Communication between people is usually affected by acoustic conditions, and sometimes the message is understood with several difficulties. There are many factors that can affect speech intelligibility; from factors inherent to the nature of the audio (e.g., resolution, signal-to-noise-ratio), to factors inherent to the nature of the voice (e.g., accent, pronunciation, tone) (Chua et al., 2017). Therefore, it is a challenge to cover the broad spectrum of such factors related to speech intelligibility.

Over the years, Speech Enhancement (SE) methods have been used to address quality issues in audio recordings. From the first (e.g., MMSE (Ephraim & Malah, 1985), WPE (Nakatani et al., 2010)) to the current state-of-the-art SE methods (e.g., TEA-PSE 3.0 (Ju et al., 2023), NAPSE (Yan et al., 2023)), they aim to suppress or mitigate the effect of noise and reverberation. Among them, there are other deep-learning-based systems that show a higher robustness to tackle nonstationary and babble noise while generating higher quality speech, such as Denoiser (Defossez et al., 2020) and MetricGAN (Fu et al., 2019). Similarly, the use of source separation methods has been adapted for speech enhancement (e.g., Hybrid Demucs (Défossez, 2021)), whose performances are comparable to the best SE models.

In recent years, the aforementioned audio quality issues have also emerged in data annotation workflows, being considered a challenge that hinders precise annotations (Springbord, 2023). Annotators often handle utterances containing different acoustic characteristics according to the environment (e.g., noise, music, etc.), the recording devices (e.g., mic type, distance, etc.), among others, becoming hard to transcribe the spoken message or annotate few sound events. Hence, noise reduction, artifact suppression, and dereverberation techniques might make the annotation process easier. Following this line, several companies in this area have built pipelines to generate large-scale high-

quality speech datasets (Liu et al., 2021), where a source separation tool was proposed as a speech enhancement method and has been proven to be effective in separating vocal and musical signals (Hennequin et al., 2020). The resulting separated vocal signal is used for further processing, tagging, and transcription to ensure higher accuracy. Regarding the importance of audio quality to obtain reliable annotations in (Cartwright et al., 2017), the authors studied how the complexity of the acoustic scene in the sound event annotation context affects the quality of the annotated events. Other works also focused on the improvement of the human transcription quality thanks to Machine Learning-in-the-loop (Gao et al., 2023).

In the context of audio annotation, it is not trivial to measure the quality of a recording to determine how it could affect the annotation and transcription task. There are several metrics that could provide a faint idea of the quality and intelligibility of a speech recording. First, many objective measures of audio quality have been proposed under the standardized ITU-R recommendations, such as the Speech Intelligibility Index (SII) (Institute, 1997) related to the listener's audibility, or the Perceptual Evaluation of Speech Quality (PESQ) (Recommendation, 2001) for telephony speech. Furthermore, Mean Opinion Score (MOS) appeared as a standardized listening test (ITU-T P.800.1 (Recommendation, 2006)) based on human ratings to evaluate the speech intelligibility from audio samples. Nevertheless, it was implemented under the domain of quality of experience and telecommunications engineering, representing the opinion about the performance of the transmission system. This process was used for both conversation and listening to spoken material (Series, 2016) but becomes time-consuming and expensive since it needs a large number of participants to carry out the evaluation. Another very well-known measure related to speech intelligibility is Word Error Rate (WER), which allows the comparison between transcriptions in the automatic speech recognition field. This measure is often used to determine the transcription quality in annotation tasks as long as a reference transcription is available.

Breakthroughs in Deep Learning have resulted in several models to predict MOS from natural speech, such as NISQA (Mittag et al., 2021) or DNSMOS (Reddy et al., 2021), even in speech synthesis, such as MOSNet (Lo et al., 2019). However, the described measures do not consider factors that could degrade annotation performance (e.g., transcription speed, annotation confidence, and the number of audio replays), since it correlates poorly with annotator perception of audio quality in the annotation context (Chua et al., 2017).

In this paper, we propose a novel subjective audio quality measure in the audio annotation framework, called the Speech Quality Score (SQS), which includes the most relevant characteristics that could degrade the transcription performance, and thus the annotation quality. In addition, motivated by the works related to MOS prediction (Mittag et al., 2021; Reddy et al., 2021; S. Oliveira et al., 2023), we investigate the use of a pre-trained model for audio quality assessment to predict SQS over a new set of annotations, which was created in this research.

This paper is organized as follows. Section 2 describes the importance of the speech quality in the annotation context, along with the dataset and the prediction model that we propose. The results are presented and discussed in Section 3. Finally, Section 4 concludes with a final summary and future works.

## 2 SPEECH QUALITY IN THE ANNOTATION CONTEXT

The naturalness of the audio (resolution, signal-to-noise-ratio) and speaker-related factors (accent, pronunciation, tone) makes it harder to get high-quality transcription. In fact, this issue is a concern of the annotation companies to develop high-quality datasets. To overcome this problem, speech enhancement methods are used as part of the annotation pipeline (Liu et al., 2021).

Nevertheless, a method to determine the speech quality of an audio sample before the annotation process would be useful to know the effort to transcribe it. Data preparation is a multi-step process where annotation and revision roles take part in it. One of the good practices is to assign specific samples according to the expertise of the team, increasing the productivity by 25% and project quality by 32% on average (SigmaAI, 2023).

In this way, more experienced annotators could receive those hard-to-annotate audio samples, even an extra service of revision could be proposed to guarantee the quality requirements. Thus, it would increase both the annotator performance and the annotation quality as it would prioritize those audio

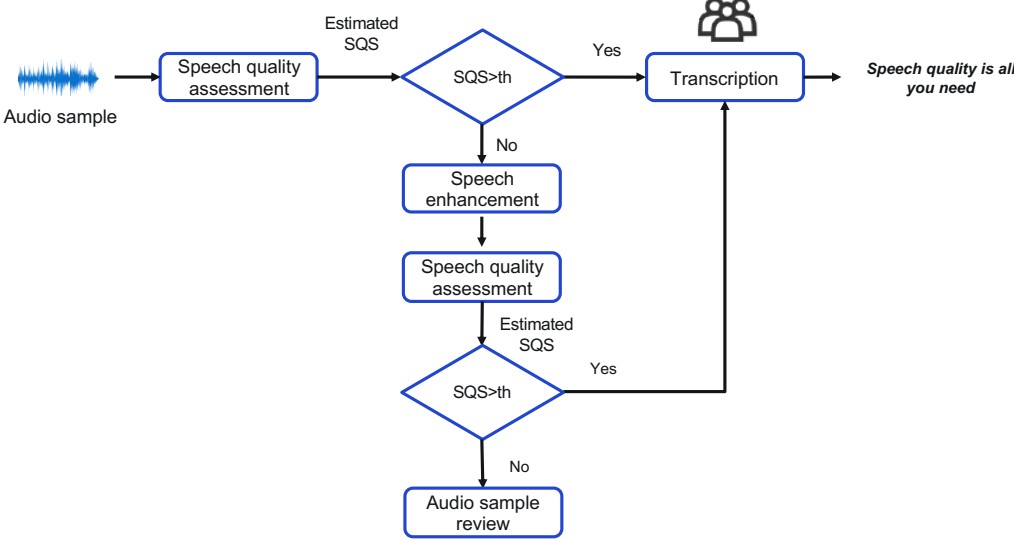

Figure 1: Scheme of speech quality assessment model as a part of the annotation pipeline. First, prior to the annotation process, the Speech Quality Score (SQS) of the input audio is assessed using the speech quality assessment method. Then it is evaluated whether it reaches the minimum speech quality set by quality thresholds (th). If the SQS is surpasses the minimum quality, the audio sample is transcribed. If not, the audio sample is processed through a speech enhancement system, and its quality is re-evaluated. Subsequently, the newly estimated speech quality is compared with the minimum speech quality (th). If the quality of the enhanced version does not reach the minimum quality, the original audio sample is forwarded to a review process.

samples with the highest quality, whereas the rest will be reviewed exhaustively in a separate process, as shown in Figure 1.

We propose a subjective measure of speech quality, which regards factors that could degrade the annotation quality; aspects that could make the transcription process difficult due to the poor quality in speech intelligibility.

Following a similar approach to MOS, we define four levels of speech quality in the annotation context (i.e., Poor, Bad, Average, Good), ranging from 1 to 4, with the lowest score of 1 (i.e., Poor) and the highest score of 4 (i.e., Good), as shown in Table 1. The purpose of such levels is to represent the difficulty of transcribing the audio due to the naturalness of the audio (i.e., background noise, channel noise) and speaker-related factors. It should be noted that these quality levels were set based on common acoustic conditions found in annotation processes, as no high-quality audio samples are expected.

## 2.1 SPEECH QUALITY EVALUATION DATA

The evaluation set consists of 1,020 audio samples in Spanish extracted from the RTVE2020 Database (Lleida et al., 2020), which contains different parts of TV programs that represent a real use case of data annotation, with a large amount of adverse acoustic conditions and challenging scenarios. The whole subset was manually selected to cover the broad spectrum of a variety of acoustic conditions from the RTVE2020 Database. Each audio sample has a duration of 10 seconds.

The objective is to define an annotation process to obtain a new set of speech quality labels regarding the annotation aspects (i.e., SQS labels) at the time of listening and transcribing the audio samples. The annotation process was carried out using a Sigma's proprietary tool for audio annotation, where each audio sample was rated by two in-house listeners, ranging from 1 to 4 according to the four quality levels, as shown in Table 1. As a result, we obtained a total of 2,040 human evaluations. In the evaluation phase, we took the average score of the two SQS ratings for each sample as the ground-truth.

Table 1: Description of the speech quality levels in terms of SQS measure.

| Quality | Description | Score |
|---|---|---|
| Poor | The speech is unintelligible. After listening more than three times to the audio, it is hard to understand. | 1 |
| Bad | The beginnings and endings are not clear or cut off; part of the speech is unintelligible. | 2 |
| Average | The audio needs to be played at least three times to understand the speech. | 3 |
| Good | The speech is mostly normal, contains a good rhythm, is easy to understand, although there is some background noise or loud | 4 |

Histograms of the mean and standard deviation are shown in Figure 2. It can be seen that the mean SQS values are concentrated around 3.5, and most of the standard deviation of the two SQS ratings is lower than 1, suggesting a high level of conciliation between annotators.

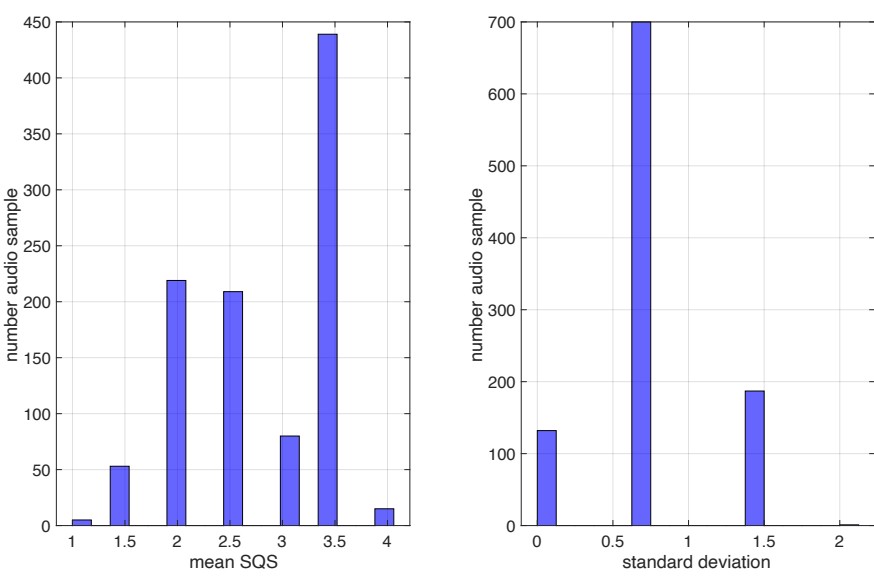

Figure 2: Histograms of the mean and standard deviation of the two SQS ratings for each audio sample.

## 2.2 SQS PREDICTION MODEL

Inspired by studies related to MOS prediction by non-intrusive methods (i.e., do not require clean reference sample to predict the quality), by using speech features extracted from audio samples and deep learning techniques to predict quality ((Mittag et al., 2021), (Reddy et al., 2021), (Yang et al., 2022)), in this paper we propose a pre-trained deep learning model as a SQS predictor. The model is the NISQA model (Mittag et al., 2021), which is an end-to-end model based on a CNN self-attention model to predict speech quality. It was trained using more than 13,000 audio samples (i.e., NISQA Corpus) with its respective subjective evaluation in terms of MOS.

Therefore, we use transfer learning to re-train the NISQA model (Mittag et al., 2021) using, as a starting point, the frozen weights[1] of the model. The audio samples are used as inputs, while their ground-truth SQS values are the outputs during the training, as depicted in Figure 3.

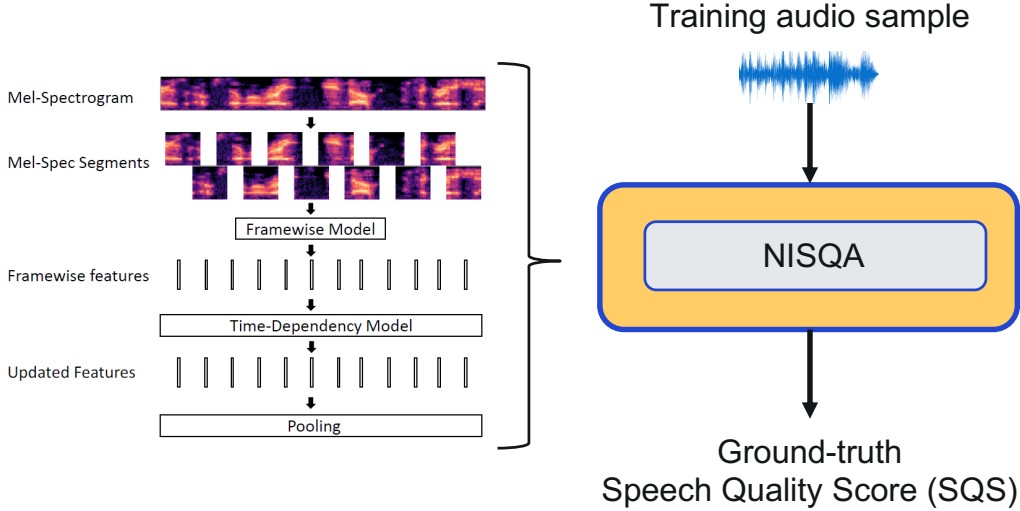

Figure 3: Scheme of the re-training process of the NISQA model for SQS metric prediction by means of transfer learning. Adapted from (Mittag et al., 2021)

## 3 EXPERIMENTS

In this section, we first analyze the relationship between the speech quality objective measures and our proposed metric, i.e., SQS. Secondly, we present the results for SQS prediction by means of the DNN-based proposed model. As evaluation metrics, we used Pearson correlation ($r$), *p-value* significance level, and Normalized Mean Square Error (NMSE).

For SQS prediction, the entire set of 1,020 speech samples along with their ground-truth SQS values were divided into 820, 113, and 87 samples for training, validation, and testing, respectively. In order to avoid false discoveries due to overfitting problems or having a small number of samples for re-training the model, the training and evaluation process were performed following the $k$-fold technique, being $k = 5$. As a result, the final performance is measured over test samples of the total number of folds.

### 3.1 COMPARISON OF OBJECTIVE SPEECH QUALITY MEASURES

We evaluate the capacity of objective speech quality measures to retain some information related to speech quality in the annotation context. In this line, we analyze the relationship between the speech objective measures and the SQS metric. Three methods for speech quality prediction were evaluated: NISQA, DNSMOS, and WER. For that, the entire set of 1,020 audio samples was used as input to the NISQA and DNSMOS models to obtain the predicted MOS. In case of WER, this measure is estimated by the comparison between the reference transcription, i.e., made by an expert team that carried out an exhaustive quality process to ensure the reliability of the transcription, and the hypothesis transcription, i.e., resulted from the audio annotation process described in Section 2.1.

As shown in Figure 4, NISQA and DNSMOS objective measures poorly correlate ($r = 0.25$) with SQS, in contrast, WER achieved a moderate correlation ($r = -0.36$). This result suggests that WER could explain part of the speech intelligibility in the annotation context as it describes the effort of

---

[1] `https://github.com/ConferencingSpeech/ConferencingSpeech2022/tree/main/baseline/weights`

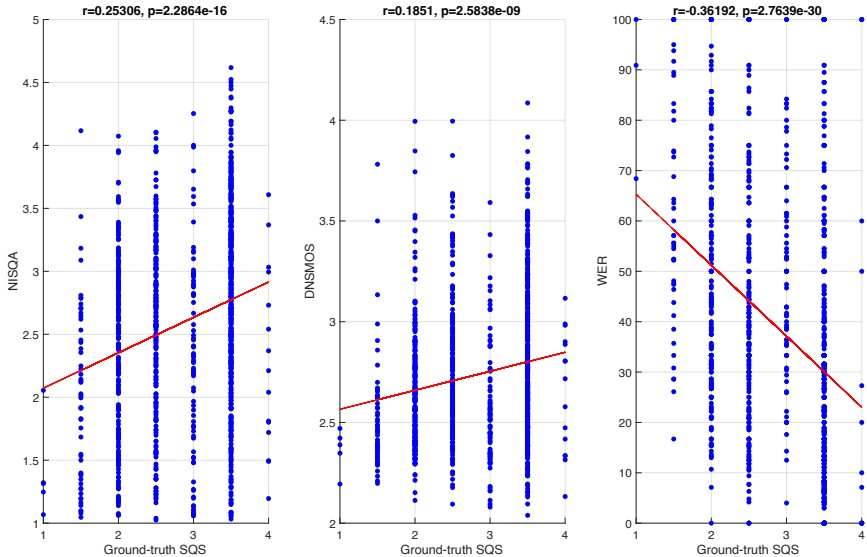

Figure 4: Scatter plot of the relationship between objective speech quality measures (NISQA, DNS-MOS, WER) and SQS for the 1,020 audio samples. The line tendency is depicted in red.

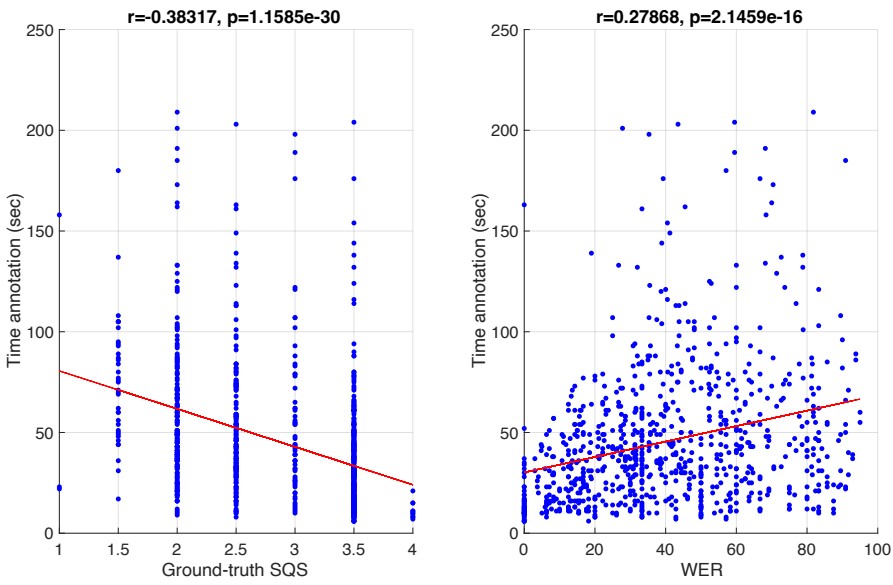

Figure 5: Scatter plot of the relationship between transcription time and SQS (left), and between transcription time and WER (right) for the 1,020 audio samples. The tendency is depicted in red.

transcribing an audio under certain acoustic conditions that could have an impact on the annotation quality. For instance, the low speech quality levels (e.g., Poor or Bad), might be related with the number of missing words or number of deletions due to the poor intelligibility of spoken content in the audio.

In addition, we assessed the impact in the annotation process, particularly the relationship between the transcription time of each utterance and both WER and SQS measures, respectively. Figure 5

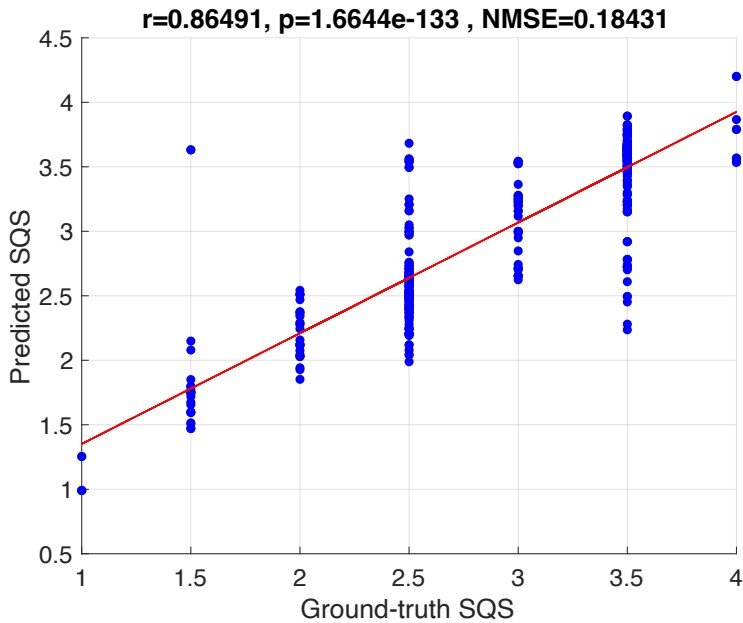

Figure 6: Scatter plot of the SQS prediction on the testing set. The line tendency is depicted in red.

Table 2: Prediction results of SQS values by the different models.

| Model | $r$ | NMSE |
|---|---|---|
| DNSMOS | 0.23 | 1.81 |
| NISQA | 0.35 | 1.09 |
| NISQA-based (Transfer learning) | 0.86 | 0.18 |

shows that annotation time and SQS maintain a higher correlation ($r = -0.38$) than the one between annotation time and WER. This result confirms that SQS captures relevant information related to the degradation of the annotation quality.

## 3.2 SPEECH QUALITY PREDICTION

Subjective speech quality in the annotation context represents the effort of transcribing an audio under certain acoustic conditions that might degrade the transcription quality, nevertheless, it is time-consuming and expensive due to the large number of participants to perform the listening test and provide perceptual ratings. In order to overcome that issue, several studies have developed DNN-based system to predict such speech quality measures, in particular the MOS, displaying high levels of accuracy in prediction of them. Following a similar approach, we evaluate the performance of the proposed system, based on the NISQA model, to predict the SQS metrics.

The results shown in Figure 6 and Table 2 confirmed the effectiveness of the proposed NISQA-based model for speech quality prediction ($r = 0.86, NMSE = 0.18$) which could be used as a substitute for human listeners to evaluate the speech quality within annotation pipeline. In contrast, the MOS prediction of testing audios samples, provided by the NISQA ($r = 0.35, NMSE = 1.09$) and DNSMOS ($r = 0.23, NMSE = 1.81$), poorly correlates with the ground-truth SQS values.

## 4 CONCLUSIONS

We present a novel subjective speech quality measure, called Speech Quality Score (SQS), suitable to be used within the audio data annotation process. This measure involves the most relevant char-

acteristics that could impact transcription performance and, consequently, annotation quality. The comparison with objective speech quality measures shows that the SQS measure includes factors that could degrade the annotation quality, showing a high correlation. In contrast, objective measures such as NISQA and DNSMOS demonstrate a weak correlation with the SQS measure. This emphasizes the necessity for an appropriate subjective measure within the annotation context, as most speech quality measures primarily address speech quality under the context of the transmission systems or the naturalness of synthetic voice.

Additionally, we applied transfer learning to adapt the NISQA model for predicting SQS metrics. Our experimental findings demonstrate that the adapted model produces yield predictions highly correlated with human ratings. This result suggests a degree of generalization of the model to predict the speech quality on a testing dataset.

As future work, validating SQS measure with other subjective measures (e.g., MOS) will confirm the relationship observed with the speech quality objective measures. Similarly, based on the capabilities of the explored systems for predicting MOS, we will explore speech representations (i.e., embeddings) based on self-supervised and supervised learning to predict SQS values.

## ACKNOWLEDGMENTS

The research leading to these results was partially supported by the EU Next Generation through the public business entity attached to the Ministry of Economic Affairs and Transformation, as a part of the 2021/C005/00146323 (HADA: Research on advanced data annotation tools). In addition, this work used data from the RTVE2020 database. This dataset was transferred by RTVE Corporation with the aim of contributing to the development of speech technologies in Spanish language.

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
