# OpenReview forum: "SQS: Speech Quality Assessment in the Data Annotation Context"
_ICLR.cc/2024/Conference — ICLR 2024 Conference Withdrawn Submission_

### Official Review · Reviewer_115n · 2023-10-19

**Soundness:** 2 fair
**Presentation:** 2 fair
**Contribution:** 1 poor
**Rating:** 3
**Confidence:** 5

**Summary:**

This paper focuses on studying the factors that may affect data annotation for audio signals. A small-scale dataset is collected (1,020 audio samples, each audio sample has a duration of 10 seconds). Then a quality estimation model is trained by transfer learning and the collected data.

**Strengths:**

The finding between transcription time and SQS, WER is interesting.

**Weaknesses:**

1) The presentation is not very clear, I’m not sure what is the main difference between the proposed Speech Quality Score (SQS) and the common MOS.
2) The novelty of this paper is very limited. In fact, there is no novelty in terms of the machine learning perspective. This paper is more suitable to be submitted to speech-related conferences (e.g., Interspeech, Icassp, etc.). Specifically, this work simply employs a pre-trained speech quality estimator (NISQA) and finetunes on its own dataset.
3) The comparison to other models (e.g., NISQA, DNSMOS) is also unfair, because of different training data and the label scale. In DNSMOS, the MOS scale is from 1 to 5, however, in the collected dataset, the label scale is from 1 to 4. Although it may be okay for Pearson correlation, metrics such as Mean Square Error (MSE) will be significantly affected by this scale mismatch.

**Questions:**

Although each audio sample is 10 seconds, do they contain the same number of words? I believe it will also affect the transcription time.

---

> ### Author Response · Authors · 2023-11-22
>
> **Weaknesses**
>
> Reviewer: The presentation is not very clear, I’m not sure what is the main difference between the proposed Speech Quality Score (SQS) and the common MOS.
>
> Reply: First, we thank the reviewer for considering our presentation fair along with a fair level of soundness. As we mention in the introduction of this paper, MOS appeared as a speech intelligibility under the domain of quality of experience and telecommunication engineering, i.e., transmission systems. However, we propose a novel speech quality measure (SQS), which cover the most relevant characteristics that could degrade the transcription performance and the annotation quality (e.g., intelligibility, number of times listening the recordings, clarity of the beginnings and endings…)
>
> Reviewer: The novelty of this paper is very limited. In fact, there is no novelty in terms of the machine learning perspective. This paper is more suitable to be submitted to speech-related conferences (e.g., Interspeech, Icassp, etc.). Specifically, this work simply employs a pre-trained speech quality estimator (NISQA) and finetunes on its own dataset.
>
> Reply: Thank you for your point of view. Despite of the fact that our work could be also suitable for speech-domain conferences, we considered this paper matching one of the ICL topic “applications in vision, audio, speech, natural language processing, robotics, neuroscience, or any other field”. We consider the novelty of this work is the definition of a metric able to provide information about how is the audio recording, i.e., related to speech intelligibility, according to the possible impact on the annotation performance, i.e., resulted label.
>
> Reviewer: The comparison to other models (e.g., NISQA, DNSMOS) is also unfair, because of different training data and the label scale. In DNSMOS, the MOS scale is from 1 to 5, however, in the collected dataset, the label scale is from 1 to 4. Although it may be okay for Pearson correlation, metrics such as Mean Square Error (MSE) will be significantly affected by this scale mismatch.
>
> Reply: We appreciate the reviewer for the observation. We agree that the scale differences between MOS and SQS values can affect MSE. In response to this, we propose using NMSE (Normalized Mean Square Error) to address these scale differences. The results with the new evaluation approach using NMSE are presented in the revised version of the paper.
>
> **Questions**
>
> Reviewer: Although each audio sample is 10 seconds, do they contain the same number of words? I believe it will also affect the transcription time.
>
> Reply: Indeed, the number of words can influence the transcription time of the file; however, in addition to that, there are other more significant features, specifically the audio quality as shown in Fig 5.

---

> > ### Comment · Reviewer_115n · 2023-12-04
> > **Response to rebuttal**
> >
> > I would like to thank the authors for their rebuttal. I suggest showing the equation/definition of Normalized Mean Square Error. I still believe simply using a pre-trained model and fin-tuned on another dataset is not novel enough for a top AI conference like ICLR. Therefore, I would like to keep my rating unchanged.

---

### Official Review · Reviewer_SKsa · 2023-11-01

**Soundness:** 3 good
**Presentation:** 2 fair
**Contribution:** 2 fair
**Rating:** 3
**Confidence:** 4

**Summary:**

This paper describes experiments to predict a new rating of the difficulty with which a speech utterance can be transcribed directly from the audio. It uses an in-house 1000-utterance dataset with this new annotation. The new annotation is correlated with NISQA predictions, DNSMOS predictions, accuracy of a single human transcriber compared to an exhaustive human transcription panel, and speed of annotating an utterance. The NISQA model can be fine tuned to predict this annotation to achieve r = 86% correlation with the ground truth.

**Strengths:**

This is an interesting application: triaging utterances for transcription by annotators of different skill level.

This seems like a well defined task that can be solved accurately with fine tuning of an existing model.

The paper is relatively easy to follow.

**Weaknesses:**

The clarity of the paper is hindered by the use of common technical terms in non-standard ways. For example, there is a frequent conflation between speech quality and speech intelligibility. The name of the proposed metric is the Speech Quality Score, although it predicts intelligibility (how easy it will be to transcribe for a human listener). While these are often related, they are not always, despite the first sentence's unsupported claim ("Speech intelligibility is directly associated with audio quality"). Similarly, MOS (Mean Opinion Score) is described as a metric even though it is merely a scale upon which many different metrics are measured (speech quality, noise suppression, overall quality, etc). Word error rate is mentioned as a measure of speech quality, but it is explicitly a measure of intelligibility. Furthermore its use in this paper is to compare one human transcription against a consensus human transcription, but this is not explained until page 5 after being mentioned several times.

The relevance of the paper to ICLR is not clear. It is a paper about speech intelligibility prediction and while of interest to the speech community, I don't think is general enough in terms of machine learning approaches or applicability to warrant publication at ICLR, which focuses on machine learning.

The reproducibility of the results is quite low without the release of either the dataset or the model. This is a new task with guidelines that could be interpreted differently from different readings of the paper, so without some aspect of the work being released, it is not clear exactly what a reader of the paper is meant to take away from it. I don't think the fact that it is possible to perform this general task is sufficiently interesting to warrant publication on its own.

It is not clear why this subjective measure of "Speech Quality Score" is necessary as opposed to a more objective measure like the actual time that it took to annotate a given utterance or the inter-rater (dis)agreement. No justification is provided, nor is any quantitative evaluation undertaken. Such an objective score would need to account for differences in utterance duration and different overall speeds between raters, so could normalize within each rater and by utterance duration. It seems that even without these normalizations, SQS is still correlated with annotation time (r=0.38). Presumably with them it would be even more correlated.

**Questions:**

Results are only provided in the paper for fine tuning the pretrained NISQA model. What are the results of training the model from scratch to predict SQS?

---

> ### Author Response · Authors · 2023-11-22
>
> **Weaknesses**
>
>  We thank the reviewer for considering our contribution fair and having a good level of soundness. In addition, we appreciate the comments about this important point to be clarified. We probably mentioned speech quality and speech intelligibility without distinction due to they are interlinked. The SQS measures audio quality, taking into account aspects or features that can impact the quality of the final transcription (e.g., background noise, number of playbacks to understand the content), and thus, the intelligibility of the transcribed content. Also as suggested by the reviewer, we have corrected the introduction of the Word Error Rate (page 2) as a measure of speech intelligibility. In the annotation context, several times the word “quality” is used to compare annotation performances, i.e., how accurate is the transcription with respect to a reference. However, we agree with the reviewer when mentions that we need to clarify this before page 5. We have re-written this part. We hope we have gained significantly in clarity.
>
> Reviewer: The relevance of the paper to ICLR is not clear. It is a paper about speech intelligibility prediction and while of interest to the speech community, I don't think is general enough in terms of machine learning approaches or applicability to warrant publication at ICLR, which focuses on machine learning.
>
> Reply: Thank you again for your point of view. Despite of the fact that our work could be also suitable for speech-domain conferences, we considered this paper matching one of the ICL topic “applications in vision, audio, speech, natural language processing, robotics, neuroscience, or any other field”. Data annotation and metrics to determine the data-quality and its impact on the annotation performance could be a general topic regardless of the nature of the data.
>
> Reviewer: The reproducibility of the results is quite low without the release of either the dataset or the model. This is a new task with guidelines that could be interpreted differently from different readings of the paper, so without some aspect of the work being released, it is not clear exactly what a reader of the paper is meant to take away from it. I don't think the fact that it is possible to perform this general task is sufficiently interesting to warrant publication on its own.
>
> Reply: The reviewer rises an interesting point, but the data from RTVE2020 dataset cannot be published because of a license agreement. In exchange, we tried to provide a detailed explanation of all the aspects regarding the data and model (Sections 2.1 and 2.2).
>
> Reviewer: It is not clear why this subjective measure of "Speech Quality Score" is necessary as opposed to a more objective measure like the actual time that it took to annotate a given utterance or the inter-rater (dis)agreement. No justification is provided, nor is any quantitative evaluation undertaken. Such an objective score would need to account for differences in utterance duration and different overall speeds between raters, so could normalize within each rater and by utterance duration. It seems that even without these normalizations, SQS is still correlated with annotation time (r=0.38). Presumably with them it would be even more correlated.
>
> Reply: We thank the reviewer for the observation. As depicted in Fig 5, the intention is to analyze how aspects related to the annotation task, such as annotation time, are better captured by the proposed metric SQS than by the objective metric WER. As mentioned before, SQS measures audio quality, taking into account aspects or features that can impact the quality of the final transcription. Moreover, SQS could be integrated within the annotation framework to predict how easy to transcribe are the utterances. The other mentioned metrics, such as the annotation time or the inter-annotator agreement, provide insights once the transcription process have been accomplished, not before.
>
> **Questions**
>
> Reviewer: Results are only provided in the paper for fine tuning the pretrained NISQA model. What are the results of training the model from scratch to predict SQS?
>
> Reply: We are really thankful for this interesting question. We have not conducted training from scratch because 1,020 samples are not sufficient to train the model. The goal in this initial exploration was to assess the potential scope that could be achieved with such a proposal.

---

> > ### Comment · Reviewer_SKsa · 2023-12-04
> > **Response to rebuttal**
> >
> > I would like to thank the authors for their rebuttal. I have considered it along with the other reviews and the authors' responses to them, but have decided to keep my review unchanged.

---

### Official Review · Reviewer_PsYv · 2023-11-02

**Soundness:** 2 fair
**Presentation:** 2 fair
**Contribution:** 1 poor
**Rating:** 5
**Confidence:** 5

**Summary:**

This paper creates a new subjective test (Speech Quality Score, SQS) for annotating speech clips. SQS is shown to correlate moderately with annotation time (r=0.38) and it is suggested to use the SQS value to determine whether to apply a speech enhancement method before transcription or not. NISQA is fine-tuned with a SQS dataset and shown to have good performance (r=0.86).

**Strengths:**

This is a novel idea that could help improve speech annotation, which is a challenging problem.
The results of the NISQA-based model are good.

**Weaknesses:**

The basic premise of the paper is SQS and the NISQA-based model can improve speech annotation (see Figure 1), but this was never done. That is, SQS and the NISQA-based model have not been shown to have any end-to-end utility.

In addition, it isn't clear SQS is even needed. Why not just do speech enhancement for all speech clips, or play both speech enhanced and originals to the annotators?

Minor issues:
Some references have errors, e.g.,
	ITUT Rec. Itu-t rec. p. 800.1 => ITU-T Rec. P.800.1
	pesq => PESQ
	Dnsmos => DNSMOS

In the introduction, these are not SOTA SE methods for denoising and dereverberation: MetricGAN (Fu et al., 2019), Denoiser (Defossezet al., 2020), SepFormer (Subakan et al., 2021). I suggest citing winners of the ICASSP DNS 2022 challenge as better examples.

**Questions:**

What is a Sigma's proprietary tool (Section 2.1)? Add a reference
Why does Table 1 not have 5: Excellent?
Why are only N=2 ratings done for the dataset in 2.1? That makes your training data fairly noisy.

---

### Official Review · Reviewer_YGuK · 2023-11-04

**Soundness:** 2 fair
**Presentation:** 2 fair
**Contribution:** 2 fair
**Rating:** 6
**Confidence:** 4

**Summary:**

The paper introduces a novel subjective speech quality measure, known as Speech Quality Score (SQS), within the audio data annotation framework. It argues that existing objective and subjective measures do not effectively consider factors that may affect the annotation process, leading to poor correlation with the audio quality perceived by the annotator. The proposed SQS measure takes into account the most relevant characteristics impacting transcription performance and, thus, annotation quality. Additionally, the authors propose a Deep Neural Network (DNN)-based model to predict the SQS measure. The experiments conducted on a dataset of 1,020 audio samples with SQS annotations show promising results.

**Strengths:**

- The paper addresses an important (and open) issue in the field of audio data annotation, highlighting the need for a more effective measure of audio quality.
- Introducing the SQS measure is innovative, considering factors directly impacting transcription performance and annotation quality.
- The use of a DNN-based model for predicting SQS metrics demonstrates a strong correlation between ground-truth and predicted SQS values, indicating the reliability of the proposed model.

**Weaknesses:**

- The paper could have expanded on the specific characteristics encompassed by the SQS measure to provide a more comprehensive understanding of its composition.
- The research relies heavily on the RTVE2020 Database for experimentation. The results might be limited and may not generalize well to other databases or real-world scenarios. For e.g., the paper does not speak much about the data collection framework setting (whether clean references provided, quality of those clean references, raters qualifications, what type of questions asked...)
- An idea of how noisy the recordings were (e.g., using a spectrogram) would have conveyed the point on how inherently noisy the recordings were (based on Fig 4(a), looks like most NISQA scores are below 3ish, so that says about the relationship b/w intelligibility and quality esp for low quality scenarios.

**Questions:**

- How can the SQS measure be validated against other subjective measures like MOS (or is MOS even the right framework for this)?
- What are the specific characteristics considered by SQS that make it more effective than existing measures? (Some ablations on combining WER and NISQA to build this hybrid metric compared to SQS might have been useful)
- Can the proposed model generalize well to other datasets or real-world scenarios?

---

> ### Author Response · Authors · 2023-11-22
>
> **********Weaknesses**********
>
> Reviewer: The basic premise of the paper is SQS and the NISQA-based model can improve speech annotation (see Figure 1), but this was never done. That is, SQS and the NISQA-based model have not been shown to have any end-to-end utility.
>
> Reply: The reviewer is right pointing out this important aspect. The aim of Figure 1 is to illustrate how important is to have a metric to determine the complexity of annotating and transcribing an audio sample, and how this metric could become useful to increase both the annotator performance and the annotation quality. In this line, our novel metric, i.e., SQS, could be integrated in the annotation pipeline to determine whether speech enhancement methods should be applied or not.
> The evaluation proposed by the reviewer is interesting; however, in this initial exploration, we aim to: 1) assess the scope of the SQS metric in a real annotation context, and 2) estimate the SQS metric using a supervised model, specifically a NISQA-based model. As future work, we will define an evaluation plan with several raters to confirm whether SQS contribute positively to the whole annotation pipeline.
>
> Reviewer: In addition, it isn't clear SQS is even needed. Why not just do speech enhancement for all speech clips, or play both speech enhanced and originals to the annotators?
>
> Reply: Thank you for the interest in this point related to the annotation process. In our experience, providing two audio recordings to the annotators could be a great idea for a small volume of audio samples. In real data annotation project, which contains millions of audio recordings, playing both original and enhanced audio supposes a low level of annotation performance, increasing the annotation time and the costs of the projects.
>
> Reviewer: Minor issues: Some references have errors, e.g., ITUT Rec. Itu-t rec. p. 800.1 => ITU-T Rec. P.800.1 pesq => PESQ Dnsmos => DNSMOS
>
> Reply: We would also like to thank the reviewer for this correction.
>
> Reviewer: In the introduction, these are not SOTA SE methods for denoising and dereverberation: MetricGAN (Fu et al., 2019), Denoiser (Defossezet al., 2020), SepFormer (Subakan et al., 2021). I suggest citing winners of the ICASSP DNS 2022 challenge as better examples.
>
> Reply: Thank you for bringing these references to our attention. We agree these works are the most relevant and well-known speech enhancement methods. We have now added the reference to these interesting papers (1st and 2nd places in the ICASSP DNS 2023 challenge):
>
> Ju, Y., Chen, J., Zhang, S., He, S., Rao, W., Zhu, W., ... & Shang, S. (2023, June). TEA-PSE 3.0: Tencent-Ethereal-Audio-Lab Personalized Speech Enhancement System For ICASSP 2023 Dns-Challenge. In ICASSP 2023-2023 IEEE International Conference on Acoustics, Speech and Signal Processing (ICASSP) (pp. 1-2). IEEE.
>
> Yan, X., Yang, Y., Guo, Z., Peng, L., & Xie, L. (2023, June). The NPU-Elevoc Personalized Speech Enhancement System for Icassp2023 DNS Challenge. In ICASSP 2023-2023 IEEE International Conference on Acoustics, Speech and Signal Processing (ICASSP) (pp. 1-2). IEEE.
>
>
> **********Questions**********
>
> Reviewer: What is a Sigma's proprietary tool (Section 2.1)? Add a reference Why does Table 1 not have 5: Excellent? Why are only N=2 ratings done for the dataset in 2.1? That makes your training data fairly noisy.
>
> Reply: We appreciate the feedback related to the whole article. Sigma’s proprietary tool refers to one of our audio annotation tools which was used to carry out the annotation process to obtain SQS values over the 1,020 audio samples. Raters were able to select a value of the defined SQS scale (1-4) for each audio sample. In Table 1, a score of 5 (Excellent) is not considered because in a real-world transcription context, the quality of the samples of the audio to be annotated often contains background noise or other artifacts, thereby affecting the intelligibility of the spoken content. In this point, the reference has been directly extracted from our clients which we cannot mention for confidentiality reasons. Furthermore, in this initial exploration, only two ratings per audio sample were considered due to resource constraints in evaluating 1,020 audios.

---

### Author Response · Authors · 2023-11-22

************Weaknesses************

We thank the reviewer for providing this general overview, it has helped us to have a better understanding of the comments below.

Reviewer: The paper could have expanded on the specific characteristics encompassed by the SQS measure to provide a more comprehensive understanding of its composition.

Reply: We agree with the reviewer that the expanded study of the different characteristics that SQS encompasses could be really interesting. For this purpose, we would have to define exhaustively the annotation process, encouraging the increase of the number of raters and planning the implementation of additional features in the annotation tool to store some statistics that are not being collected right now.

Reviewer: The research relies heavily on the RTVE2020 Database for experimentation. The results might be limited and may not generalize well to other databases or real-world scenarios. For e.g., the paper does not speak much about the data collection framework setting (whether clean references provided, quality of those clean references, raters qualifications, what type of questions asked...)

Reply: Thank you for the observation related to the datasets. RTVE2020 dataset was chosen for this purpose because it contains TV programs that represent a real-world scenario of data annotation. In addition, it is important to highlight that the RTVE2020 dataset has been used to validate by other studies regarding speech technologies in Spanish. Coming back to the core of the matter, companies in the multimedia sector often require data annotation tasks for media content indexing and searches on them, among others. Audio samples were directly obtained from this dataset. Our proprietary annotation tool for audio annotation provides the audio samples (16 kHz) to listeners along with a set of values (Table 1) to assign one of these values to each audio sample.

Reviewer: An idea of how noisy the recordings were (e.g., using a spectrogram) would have conveyed the point on how inherently noisy the recordings were (based on Fig 4(a), looks like most NISQA scores are below 3ish, so that says about the relationship b/w intelligibility and quality esp for low quality scenarios.

Reply: We thank the reviewer for the suggestion. We agree that the spectrogram can help understand intelligibility in low-quality scenarios.

************Questions************

Reviewer: How can the SQS measure be validated against other subjective measures like MOS (or is MOS even the right framework for this)?

Reply: We intend to validate our subjective SQS measure by analyzing its correlation with another subjective quality measure, such as MOS (Mean Opinion Score). To achieve this, we propose to assess the quality of the samples in terms of MOS and subsequently calculate correlations between both subjective measures. In this way, we would confirm the correlations found between SQS and MOS (Figure 4).

Reviewer: What are the specific characteristics considered by SQS that make it more effective than existing measures? (Some ablations on combining WER and NISQA to build this hybrid metric compared to SQS might have been useful)

Reply: One of the most relevant features of SQS is that it captures audio quality in an annotation context (Table 1), unlike WER and NISQA (i.e., MOS estimation). In the case of WER, it represents intelligibility but not audio quality directly. For example, low WER (10%) or high WER (40%) does not provide much information about audio quality or how it may impact the difficulty of the transcription task.

Reviewer: Can the proposed model generalize well to other datasets or real-world scenarios?

Reply: That is one of the aspects that we would like to assess, as in this first part, the evaluation has only been conducted on the RTVE 2020 dataset. However, as mentioned before, SQS measures were indeed obtained through a real annotation pipeline, meaning that expert annotators and a proprietary annotation tool were employed.

---

### Meta-Review · Area_Chair_GLgJ · 2023-12-06

**Metareview:**

The authors propose a new speech quality metric, Speech Quality Score (SQS), and a DNN based model to predict SQS. The measure is defined based on annotation descriptions and divided into the range [1, 4]. The authors then adapt a pre-trained DNN to predict it using the small dataset they collect for the purpose. The results show that SQS correlates with WER< and that predictions are reasonably accurate.

The study is too small-scale at the moment. The training set is constructed using just ~800 utterances that were 2-way annotated. It is not surprising that a DNN trained on matched data gives good quality. The SQS metric correlating with WER is probably also a factor of how it’s defined and annotated. It is interesting to see that measures like NISQA and DNSMOS don’t correlate with WER, which points to the need for better metrics.

Reviewers brought a few issues like lacking description on the characteristics of SQS, and a lack of comprehensive understanding of the metric (how does it correlate with other subjective metrics like MOS, what exactly does it capture, etc.). The small-scale nature is also a serious  limitation, and it’s unclear if it will generalize to other data. Other issues also exist, like claims about being useful for annotation is not validated, missing comparisons with other approaches like MetricGAN. It was also pointed out that the paper is perhaps not ideal for an ICLR audience.

**Justification For Why Not Higher Score:**

The small-scale nature of the work, insufficient experimentation / analysis of the proposed metric are significant limitations at the moment.

**Justification For Why Not Lower Score:**

N/A

---

### Decision · Program_Chairs · 2024-01-16

Reject